Effects of chemical surface pretreatments on the surface roughness and shear bond strength of fiber post-resin cement interfaces: a preliminary in vitro study

http://orcid.org/0000-0001-6199-0472 Turkes Basaran Elif 1 elif.turkes@yeditepe.edu.tr
Kazak Magrur 2
Gokce Kagan 1
Benderli Gokce Yasemin 3
1 Department of Restorative Dentistry, Faculty of Dentistry, Yeditepe University , Istanbul , Turkey
2 Department of Restorative Dentistry, School of Dental Medicine, Bahcesehir University , Istanbul , Turkey
3 Department of Restorative Dentistry, Faculty of Dentistry, Istanbul University , Istanbul , Turkey
Farooq Imran
Electronic publication date: 2025 Aug 11
Publication date: 2025
Volume: 13
Electronic Location ID: e19853
Received 2025 Mar 6; Accepted 2025 Jul 15
Copyright: © 2025 Turkes Basaran et al.
Copyright year: 2025
Copyright holder: Turkes Basaran et al.
License: This is an open access article distributed under the terms of the Creative Commons Attribution License, which permits unrestricted use, distribution, reproduction and adaptation in any medium and for any purpose provided that it is properly attributed. For attribution, the original author(s), title, publication source (PeerJ) and either DOI or URL of the article must be cited.
License URL: https://creativecommons.org/licenses/by/4.0/

Keywords: Fiber post, Hydrofluoric acid, Hydrogen peroxide, Shear bond strength, Surface roughness

Funding: The authors received no funding for this work.

==============================
Background

This preliminary study evaluates the impact of different chemical surface pretreatments on the surface roughness (Ra) of fiber posts and the shear bond strength (SBS) at the interface between the horizontal surface of fiber posts and resin cement, highlighting an area that has not been extensively researched.

Methods

Fifty glass fiber posts (Cytec Blanco, Hahnenkratt) were used, and polishing discs were employed to create a horizontally flat surface. Their dimensions were standardized using a digital caliper and divided into two pieces from each post (n = 20). The experimental groups included different pretreatment protocols: Group I (Control) had no pretreatment, Group II used 4% hydrofluoric acid (HF) (Porcelain Etchant, Bisco), Group III used 9.6% HF (Porcelain Etch Gel, Pulpdent), Group IV used 20% hydrogen peroxide (HP) (Merck KGaA), and Group V used 35% HP (Whiteness HP, FGM). Ten specimens from each pretreatment group were used for Ra evaluation (Perthometer Mahr), while the other ten were prepared for SBS testing. Silane and primer were applied before dual-cure resin cement (Multilink N, Ivoclar Vivadent), and SBS was tested after 48 h using a universal testing machine (Instron). SBS and Ra data were analyzed with one-way ANOVA and Tukey’s post hoc tests (p < 0.05).

Results

Statistical analysis revealed no significant pairwise differences when evaluating SBS values (p = 0.055). However, Groups IV and V exhibited higher SBS values compared to Groups I, II, and III. While 20% HP and 4% HF did not significantly enhance SBS, higher concentrations of HP achieved clinically acceptable bond strength. The surface pretreatment methods significantly influenced Ra (p = 0.00003). Groups III and IV exhibited significantly higher Ra values than Groups I, II, and V.

Conclusion

This preliminary study indicates that chemical surface pretreatments can be safely used to enhance the bond strength between fiber posts and resin cement, as they are not time-consuming, easy to use, and clinically applicable.

Introduction

Endodontic treatment is required due to deep carious lesions, trauma, wear, or failures from previous restorations. After endodontic treatment, significant dental tissue loss usually weakens the dental structure. Meanwhile, the physical and mechanical properties of the tooth, as well as the aesthetic features of the remaining dental tissue, undergo changes (Ferrari & Scotti, 2002). Therefore, it is necessary to restore the aesthetic and function of teeth that have undergone root canal treatment. Today, alongside many treatment options, post systems remain widely used for treating endodontically treated teeth with extensive tissue loss (Akkayan & Gulmez, 2002). Using a post as a treatment option ensures adequate restoration and crown-root continuity, thereby restoring the tooth’s function. Fiber posts are currently regarded as an excellent alternative to cast metal posts. The elastic modulus of fiber posts is similar to that of dentin tissue, which provides appropriate stress distribution in endodontically treated teeth (Aksornmuang et al., 2004; Monticelli et al., 2006b).

Fiber posts available for commercial purchase consist of unidirectional fibers, such as carbon, quartz, or glass, that are embedded in a resin matrix. This resin matrix provides compression resistance while also shaping the post (Mannocci, Sherriff & Watson, 2001).

Concerning previous studies, the design, length, diameter, and type of surface treatment of a post are factors that affect its attachment. The failure of glass fiber posts is usually due to the separation of bonds at the post-cement interface, which is a result of weak bonding. Therefore, various post-surface treatment procedures have been explored to improve the bond strength between the fiber post and the resin cement (Monticelli, Ferrari & Toledano, 2008; Yenisey & Kulunk, 2008). Recently suggested post-surface treatment procedures in the literature include silanization, mechanical pretreatments (sandblasting with aluminum oxide or silica particles), and chemical etching with solutions (hydrofluoric acid, hydrogen peroxide, potassium permanganate) (Mazzitelli et al., 2008; Monticelli et al., 2006b; Monticelli, Ferrari & Toledano, 2008; Souza et al., 2022). Several researchers have investigated the use of silane coupling agents to enhance adhesion between fiber posts and resin cement. However, there is still no definitive agreement on this issue (Mazzitelli et al., 2008; Moraes et al., 2015). To create a strong bond between the fiber post and resin cement, silanization alone is not sufficient. Therefore, mechanical pretreatment methods and chemical solution applications have also been utilized. These methods aim to remove the resin-based surface layer of fiber posts, preparing the fibers for silanization (Mazzitelli et al., 2012).

When the effects of the mechanical pretreatment methods on the bond strength between fiber posts and resin cement were evaluated, it was found that mechanical pretreatments effectively increased the bond strength (O’Keefe, Miller & Powers, 2000). In addition, the researchers noted that despite the good bond strengths achieved with mechanical pretreatments, the shape of the fiber posts could be disrupted, resulting in problems with their placement in root canals (Aksornmuang et al., 2004). Conversely, chemical solutions also aim to increase the micromechanical adhesion between the post and the resin cement by roughening the post surface (Monticelli, Ferrari & Toledano, 2008).

This in vitro study aimed to assess and compare the effects of various chemical surface pretreatments on the surface roughness (Ra) of fiber posts, as well as the shear bond strength (SBS) between the prepared horizontal surface of the glass fiber post and resin cement. This preliminary study was necessary due to the innovative clinical application, which has not yet been researched. The null hypotheses of this study were: (1) Surface pretreatments applied to glass fiber posts using different concentrations of various chemical solutions do not affect shear bond strength at the post-resin cement interface. (2) Different surface pretreatment methods do not influence the surface roughness of the fiber post.

Materials and Methods

The Yeditepe University Faculty of Dentistry Scientific Committee approved this investigation with No. 393.

The materials used in this study are shown in Table 1.

Table 1 Commercial names, manufacturers, lot numbers, and compositions of the materials used.

Experimental materials (commercial name)	Manufacturer	Lot number	Composition	
Cytec Blanco	Hahnenkratt, Germany	37176	60% glass fiber, 40% epoxy resin matrix
(Yenisey & Kulunk, 2008)	
Porcelain Etchant	Bisco, USA	1900006159	4% hydrofluoric acid (Yoo et al., 2015)	
Porcelain Etch gel	Pulpdent, USA	1315100849	9.6% hydrofluoric acid
(Güler et al., 2006)	
Whiteness HP	FGM, Brasil	170815	35% hydrogen peroxide
(Menezes et al., 2014)	
Hydrogen peroxide 30% (Perhydrol)	Merck KGaA, Germany	1072091000	30% hydrogen peroxide	
Monobond N	Ivoclar Vivadent, Liechtenstein	V43822	Ethanol, water, %1wt 3-methacyloxypropyltrimethoxysilane
(3-MPS)
(Monticelli et al., 2006b)	
Multilink N	Ivoclar Vivadent, Liechtenstein	V35943	Monomer matrix: Dimethacrylate and HEMA
Inorganic fillers: Barium glass, ytterbium trifluoride, spheroid mixed oxide
(Aleisa et al., 2013)	

Fifty glass fiber posts (Cytec Blanco, Hahnenkratt, Konigsbach-Stein, Germany) were used, each measuring 1.9 mm in diameter and 20 mm in length. Before starting the experiment, parallel posts each measuring 13 mm in length were obtained by cutting 7 mm tapered apical sections with a diamond saw under water cooling. After obtaining the parallel posts, they were fixed horizontally to a flat surface using double-sided adhesive tape. The horizontal surfaces of the posts were then adjusted sequentially with polishing discs (OptiDisc, Kerr, CA, USA) to achieve a flat surface. First, the horizontal side of the post was flattened until the thickness reached 1.4 mm to ensure both sides were equally flat. The post was then rotated 180 degrees, and the bottom side was flattened to a thickness of 0.8 mm for easy application. The remaining post thicknesses were measured throughout the process with a digital caliper. Finally, each fiber post was cut in half with a diamond saw under water cooling to obtain 50 samples (Fig. 1).

Figure 1 Illustration of the post-preparation and horizontal post surface.

Before pretreatment, the posts were cleaned using 96% ethyl alcohol for 3 min, then rinsing with distilled water and air dried. Hydrofluoric acid (HF) and hydrogen peroxide (HP) were used for surface pretreatments at different concentrations.

Based on the type of surface pretreatment, the samples were divided into five groups, each containing 10 samples for SBS evaluation and 10 samples for Ra evaluation. The sample size was determined for SBS and Ra tests through power analysis using G*Power 3.1 software (Dusseldorf, Germany), aiming for a statistical power of 80% and a significance level of 0.05. The effect sizes (d) were calculated using the mean values obtained from groups with equal sample numbers in similar studies of SBS (Mazzitelli et al., 2008) and Ra tests (Sharma et al., 2014). As a result of the calculation performed using these values, n = 10 was taken for each SBS and Ra evaluation.

Experimental and control groups of the study

Group I: No surface pretreatment was performed (Control).

Group II: A 4% solution of hydrofluoric acid (Porcelain Etchant, Bisco, Schaumburg, IL, USA) was uniformly applied to the surface of the post for a duration of 1 min. This was subsequently followed by thorough rinsing with distilled water, accompanied by gentle air drying to ensure optimal results.

Group III: The 9.6% hydrofluoric acid (Porcelain Etch Gel, Pulpdent, Watertown, MA, USA) was applied on the post surface for 15 s, followed by washing and rinsing with distilled water and gentle air drying.

Group IV: The specimens were immersed in 20% hydrogen peroxide solution, diluted from 30% HP solution (Merck KGaA, Darmstadt, Germany), for 20 min at room temperature, followed by washing and rinsing with distilled water and gentle air drying.

Group V: The specimens were immersed in 35% hydrogen peroxide (Whiteness HP; FGM, Joinville, SC, Brazil) solution for 1 min, followed by washing and rinsing with distilled water and gentle air drying.

For SBS evaluation, the samples were randomly divided into five groups according to the surface pretreatment methods (n = 10). After the surface pretreatments, a silane coupling agent (Monobond N, Ivoclar Vivadent AG, Schaan, Liechtenstein) was actively applied to the post surfaces with a micro brush and left for 60 s according to the manufacturer’s instructions. Then, the dual-cure resin cement (Multilink N; Ivoclar Vivadent AG, Schaan, Liechtenstein) was applied to the treated post surfaces using a polytetrafluoroethylene round mold (1.5 mm hole diameter, 4 mm thickness) and polymerized with an LED light-curing unit (Demi Ultra; Kerr Corp. Orange, CA, USA) with an output of 1,100 mW/cm2 for 20 s. Samples were embedded horizontally in the center of an auto-polymerizing acrylic resin mold (Fig. 2) and stored in distilled water at 37 °C for 48 h. The shear bond strength of each specimen was measured using a universal testing machine (Instron 3345; Instron Corp., Canton, MA, USA) at a crosshead speed of 1 mm/min. (DIN 13990 (2017)) (Fig. 3). Shear bond strength was calculated from the failure Newton (N) values in terms of megapascal (MPa) with a bonding area for all specimens.

Figure 2 Illustration of dual-cure resin cement application on the horizontal fiber post surface by a round plastic mold.

Figure 3 Illustration of the fiber post sample on the universal testing device.

The failures were examined carefully under a microscope (Carl Zeiss Meditec AG, Jena, Germany) at 15× magnification. The examination aimed to determine the type of failure and classify it as one of the following: adhesive failure between the post and cement, cohesive failure within the post, cohesive failure within the cement, or a mix of both.

The remaining samples for Ra evaluation were randomly divided into five groups (n = 10) and prepared as previously described. Using a contact-type profilometer (Perthometer M1; Mahr, Göttingen, Germany), each sample’s surface roughness (Ra) was determined. The device’s probe moved at a speed of 2 mm/s, and measurements of the samples’ surface roughness were taken from three distinct sites at intervals of 2 mm. The average Ra value (Ra, μm) was then calculated using the arithmetic mean. Before measuring each group, the profilometer was calibrated. One specimen from each group close to the mean Ra value was selected for scanning electron microscopy (SEM) analysis (6335-F; JEOL Ltd., Tokyo, Japan).

Descriptive statistics were computed for the groups. The Kolmogorov–Smirnov test confirmed the normality of SBS and Ra samples. One-way ANOVA and the post hoc Tukey HSD tests were performed (SPSS Software version 26; IBM, Armonk, NY, USA) (p < 0.05).

Results

Table 2 shows the mean shear bond strength (SBS), standard deviation values, and pairwise comparisons of the experimental and control groups regarding SBS for all experimental groups. When comparing each pretreatment group with the control group, no statistically significant difference was observed between them (p > 0.05). Additionally, no statistically significant difference was found among the pretreatment groups (Tables 2 and 3) (p > 0.05).

Table 2 Pairwise comparisons of experimental and control groups in terms of shear bond strength (MPa).

Groups	Mean ± Standard. deviation (MPa)	p *	
Control group	16.83 ± 3.55	0.983	
4% HF	17.98 ± 2.84		
Control group	16.83 ± 3.55	0.849	
20% HP	19.01 ± 3.66		
Control group	16.83 ± 3.55	0.064	
35% HP	22.73 ± 6.36		
Control group	16.83 ± 3.55	0.251	
9.6% HF	21.28 ± 6.42		
4% HF	17.98 ± 2.84	0.989	
20% HP	19.01 ± 3.66		
4% HF	17.98 ± 2.84	0.196	
35% HP	22.73 ± 6.36		
4% HF	17.98 ± 2.84	0.545	
9.6% HF	21.28 ± 6.42		
20% HP	19.01 ± 3.66	0.427	
35% HP	22.73 ± 6.36		
20% HP	19.01 ± 3.66	0.826	
9.6% HF	21.28 ± 6.42		
35% HP	22.73 ± 6.36	0.962	
9.6% HF	21.28 ± 6.42		
Notes:

* Post hoc Tukey HSD test.

p < 0.05.

Table 3 Summary of one-way ANOVA showing the effect of different surface pretreatments on SBS of fiber post-to-resin cement (MPa).

ANOVA	
SBS	Sum of squares	df	Mean square	F	Sig.	
Between groups	232.702	4	58.176	2.512	0.055	
Within groups	1,042.205	45	23.160			
Total	1,274.907	49				

When the study results were evaluated, the Control group (Group I) exhibited the lowest SBS values numerically (16.83 ± 3.55 MPa). The 4% HF group (Group II) demonstrated higher values (17.98 ± 2.84 MPa) than the Control group but lower than the 20% HP group (Group III) (19.01 ± 3.66 MPa). The high-concentration groups, 35% HP and 9.6% HF (Groups IV and V), displayed higher values than the low-concentration groups (Groups II and III). While the highest SBS values were found in the 35% HP (22.73 ± 6.36 MPa) and 9.6% HF groups (21.28 ± 6.42 MPa), respectively, no statistically significant difference was observed between these groups (p > 0.05).

After SBS testing, fracture modes were examined under a microscope (15× magnification). Only one type of fracture mode was observed, which was cohesive in the cement material (Mosharraf & Yazdi, 2012) (Fig. 4).

Figure 4 Microscopic image of the failure mode.

Table 4 displays the mean surface roughness (Ra), standard deviation values, and pairwise comparisons for all experimental and control groups. The various surface pretreatment methods significantly influenced Ra (p = 0.00003; Tables 4, 5). Groups III and IV exhibited significantly higher Ra than Group I, II, and V, while no significant differences were observed among Groups I, II, and V or between Groups III and IV (p > 0.05, Table 4).

Table 4 Pairwise comparisons of experimental and control groups in terms of surface roughness (Ra; μm).

Groups	Mean ± Standard deviation (μm)	p *	
Control group	0.6 ± 0.12	0.762	
4% HF	0.51 ± 0.09		
Control group	0.6 ± 0.12	0.015	
20% HP	0.85 ± 0.18		
Control group	0.6 ± 0.12	0.999	
35% HP	0.58 ± 0.25		
Control group	0.6 ± 0.12	0.026	
9.6% HF	0.83 ± 0.14		
4% HF	0.51 ± 0.09	0.0001	
20% HP	0.85 ± 0.18		
4% HF	0.51 ± 0.09	0.879	
35% HP	0.58 ± 0.25		
4% HF	0.51 ± 0.09	0.001	
9.6% HF	0.83 ± 0.14		
20% HP	0.85 ± 0.18	0.008	
35% HP	0.58 ± 0.25		
20% HP	0.85 ± 0.18	1	
9.6% HF	0.83 ± 0.14		
35% HP	0.58 ± 0.25	0.014	
9.6% HF	0.83 ± 0.14		
Notes:

* Post hoc Tukey HSD test.

p < 0.05.

Table 5 Summary of one-way ANOVA showing the effect of different surface pretreatments on the fiber posts’ surface roughness (μm).

ANOVA	
Surface roughness	Sum of squares	df	Mean square	F	Sig.	
Between groups	0.944	4	0.236	8.507	0.000	
Within groups	1.249	45	0.028			
Total	2.193	49				

SEM observations of the samples in Group I, Group II, and Group V (Figs. 5, 6, 7 respectively) exhibited smoother and relatively similar surface morphologies compared to Groups III and IV (Figs. 8, 9). Group IV revealed a rough surface with exposed fibers (Fig. 9), whereas Group III showed observable dissolution of the organic matrix and fiber deterioration (Fig. 8).

Figure 5 SEM image of the fiber post with no treatment.

Figure 6 SEM image of the fiber post pretreated with 4% hydrofluoric acid.

Figure 7 SEM image of the fiber post pretreated with 35% hydrogen peroxide.

Figure 8 SEM image of the fiber post pretreated with 9.6% hydrofluoric acid.

Figure 9 SEM image of the fiber post pretreated with 20% hydrogen peroxide.

Discussion

In glass fiber posts, the fibers consist of glass, and the resin matrix is epoxy resin. Since the epoxy polymers surrounding the fibers have a cross-linked structure, they cannot form strong bonds with resin structures (Mosharraf & Ranjbarian, 2013). Failures of fiber posts in clinical conditions are attributed to cementation at the fiber post-cement interface (Kim et al., 2013; Vano et al., 2006). The glass fibers within the post must be exposed for chemical interaction to occur at the fiber post-resin cement interface (Yenisey & Kulunk, 2008). To achieve this, researchers suggest various methods such as mechanical pretreatment methods (sandblasting with aluminum oxide or silica particles) and chemical solution applications to the post surface to increase the bond strength between the fiber post and the resin cement (Monticelli, Ferrari & Toledano, 2008).

Studies investigating different surface pretreatments on fiber posts report that mechanical surface treatments can cause structural deterioration of the post surface. Due to the large surface indentations after mechanical pretreatments, water may infiltrate the post-cement interface, leading to hydrolytic degradation and a weakening of the bond (Ferracane, 2006). Although mechanical surface applications initially yield high bond strength results, studies show that the bond weakens over time (Machry et al., 2020). For this reason, chemical solutions were applied for the glass fiber post-surface pretreatments in this study.

Various chemical solutions (such as hydrogen peroxide, hydrofluoric acid, potassium permanganate, etc.) for etching the post surface have been used to evaluate the bond strength between the fiber post and resin cement. The preferred hydrofluoric acid and hydrogen peroxide solutions in this study were chosen because they are not time-consuming, easy to use, clinically applicable methods to enhance the bond strength between fiber posts and resin cement, and they do not cause discoloration, unlike potassium permanganate (Mazzitelli et al., 2008; Roizard, Wery & Kirmann, 2002).

Hydrofluoric acid is primarily used for etching dental glass ceramics to enhance the micromechanical interaction with resin cement (Knotter, 2000). This acid was recently suggested for etching glass fiber posts (D’Arcangelo et al., 2007). Etching with HF acid roughens the post surface, promoting micromechanical coupling with the resin cement. It has been stated that the effect of HF acid is influenced by time, concentration, and the composition of the post, including the type of matrix and fibers (Monticelli et al., 2008; Sipahi et al., 2014). In some studies where HF acid was used as a surface pretreatment, it was reported that high acid concentration contributed to the bond strength (Sipahi et al., 2014; Vano et al., 2006). Conversely, other studies indicated that it altered the surface topography of the post to a degree similar to mechanical surface treatments (Belwalkar, Gade & Mankar, 2016; Naves et al., 2011; Sharma et al., 2014). Given the varying data from these studies, both high (9.6%) and low (4%) concentrations of HF acid were selected for this study to reveal and compare their actual effects on the post surface.

Hydrogen peroxide (HP) is a chemical solution known for its effectiveness in dissolving the epoxy resin encasing fiber posts. Research indicates that varying concentrations of HP can partially dissolve the epoxy resin, thereby exposing the underlying glass fibers of the post. This action facilitates the creation of a micromechanical retention area for resin cement by removing a superficial layer of the epoxy resin. Furthermore, it allows for the exposure of glass fibers, enabling chemical bonding with the silane agent, all while preserving the integrity of the post surface (Naves et al., 2011; de Sousa Menezes et al., 2011). Vano et al. (2006) conducted a study in which they used HF and HP as surface pretreatments to evaluate the bond strength between fiber posts and various resin composites. They reported that applying HP to the post surface contributed to the bond strength. Various concentrations of the solution have been evaluated in the studies (Cadore-Rodrigues et al., 2020; Naves et al., 2011; de Sousa Menezes et al., 2011). It has been reported that low percentages of HP solutions, such as 6% and 10%, may not be sufficient to dissolve the epoxy resin surface. Therefore, the present study evaluated two different concentrations of HP solutions (20% and 35%). The 20% HP solution was obtained by diluting the 30% HP solution, and the application time was determined to be 20 min (Mosharraf & Ranjbarian, 2013). Considering the availability and application times of these solutions, their use in clinical conditions may pose challenges. Therefore, a 35% HP solution, which is easy to obtain and apply, was included in the study.

With chemical or mechanical surface pretreatments, more fibers can be exposed on the post surface, allowing for greater interaction. The bond strength may be enhanced by providing chemical interaction and micromechanical bonding between the exposed silanized fibers and resin materials (Belwalkar, Gade & Mankar, 2016; Naves et al., 2011). Numerous studies have demonstrated the effectiveness of silane in improving the adhesion of resin materials to fiber posts (Moraes et al., 2015). These findings are based on silane’s ability to enhance surface wettability, leading to the formation of chemical bridges with OH-covered substrates such as glass or quartz fibers. Only through silanization can this chemical interaction occur between the composite resin and the exposed glass fibers of the post (Monticelli, Ferrari & Toledano, 2008). Consequently, silane was applied to the surface of the fiber post in this study, both in the control group and following surface treatments.

In clinical applications, cement is in contact with the horizontal surfaces of the post. However, no study has tested the shear bond strength from the horizontal side surface of the post. In contrast, all the literature focuses on evaluating the bond strength of the fiber post and resin cement interface using shear tests on the top parts of the posts. Nonetheless, the experimental setup does not accurately simulate clinical applications. Therefore, innovative clinical application of the horizontal surface of the fiber post was used to evaluate the SBS in this study.

Shear bond strength tests are widely used to evaluate the interface bond strength. In studies examining the fiber post-resin cement interface shear bond strength, it was consistently found that the experiments were set up with only the upper part of the post surface interacting with the resin cement (Reza & Ibrahim, 2015; Yenisey & Kulunk, 2008). However, the clinical situations, when applying the post material in the root canal, the horizontal surfaces of the fiber posts also interact with the adhesive agent and the cement. For this reason, in this in vitro study, to accurately reflect the clinical scenario, the horizontal surfaces of the fiber posts were prepared, resin cement was applied to these surfaces, and the shear bond strength between the horizontally prepared post surface and resin cement was evaluated.

According to this in vitro study, the first null hypothesis has been accepted because there were no statistical differences between the groups, despite the numerical improvements achieved with the increased percentage of surface pretreatments. On the other hand, the second null hypothesis was rejected, indicating that the surface pretreatments affected the surface roughness of the fiber posts.

According to studies by Vano et al. (2006) and Monticelli et al. (2006a), oxidizing solutions like hydrogen peroxide and hydrofluoric acid can selectively dissolve the epoxy resin covering fibers. Belwalkar, Gade & Mankar (2016) also observed this phenomenon through microscopic analysis when they used 4% hydrofluoric acid to prepare fiber posts. By selectively dissolving the glass component of the fiber post, an irregular pattern of microcracks was created, which enhanced the penetration of the resin (Belwalkar, Gade & Mankar, 2016). Gencoglu et al. (2013) reported that hydrofluoric acid (9.6%) treatment was ineffective in studies evaluating the bonding strength between fiber post and resin cement. On the other hand, D’Arcangelo et al. (2007) found that hydrofluoric acid treatment (9.5%) enhanced the bond strength of the post. In this study, HF acid of different concentrations did not statistically affect the shear bond strength values between the fiber post and resin cement (p > 0.05). However, HF acid with a higher concentration (9.6%) showed higher shear bond strength values, above the clinically acceptable value (Hemadri et al., 2014; Zakavi et al., 2021). In the low-concentration hydrofluoric acid group, similar roughness was observed compared to the Control group, and the surface exhibited a cleaner appearance (Figs. 5, 6). The high-concentration hydrofluoric acid group showed higher surface roughness values than the other groups, except the 20% HP group. SEM image also reveals surface deformation and fiber exposure (Fig. 8). HF acid with a higher concentration may have increased the SBS by enhancing the surface roughness.

In contrast, hydrogen peroxide exhibits a gentler nature, as it preserves the smoothness of the exposed fibers while ensuring that the underlying epoxy resin matrix remains intact following the etching treatments (Belwalkar, Gade & Mankar, 2016; Monticelli et al., 2006a; Roizard, Wery & Kirmann, 2002). Monticelli et al. (2006a) explained that using hydrogen peroxide for etching is an uncomplicated, efficient, and practical method to enhance the bonding between fiber posts and resin composites in a clinical setting, without requiring corrosive chemicals.

Naves et al. (2011) examined the effects of surface modification on glass fiber posts. The posts were treated with 10% hydrogen peroxide for 20 min and 24% hydrogen peroxide for 10 min, followed by the application of a silane coupling agent. The results demonstrated that this treatment improved the bond strength between the post and resin. When other studies comparing these two concentrations were evaluated, it was observed that the bonding values increased after applying solutions with higher concentrations (Valdivia et al., 2014; Vano et al., 2006). According to a recent study, using a concentration of 35% hydrogen peroxide (HP) results in better oxidation and bonding strength than a 24% concentration, regardless of the application method or immersion (Menezes et al., 2014). Although there was no statistical difference between the SBS values of HP solutions in this study, increased SBS values were obtained after applying a higher concentration of HP solution, paralleling the results of other studies mentioned. The low-concentration hydrogen peroxide group showed higher roughness values than the high-concentration hydrogen peroxide group. This may be due to the longer application time in the low-concentration hydrogen peroxide group. No deformation was observed in the high-concentration hydrogen peroxide group (Fig. 7), and the surface roughness was similar to that of the Control group.

However, the roughness values obtained did not correlate with the SBS values. This may be due to the narrow and small area of fiber applied, which prevents the effect from being fully observed.

In vitro studies assessing the bond strengths of fiber posts and resin cement also explore fracture types. Failure modes suggest that materials with high SBSs will exhibit cohesive failure within the material. Conversely, materials with low bond strengths may display more adhesive failures than cohesive ones (Bouschlicher, Reinhardt & Vargas, 1997). Thus, in a study of failure modes, failures occurring within the material (cohesive failure) can be more advantageous for withstanding forces. In the present study, a cohesive fracture mode within the cement material was noted, which is favorable (Fig. 4).

In the present study, the shear bond strength values between the fiber post and the resin cement were not statistically affected by the different concentrations of the various surface pretreatments (HF and HP) applied to the post surfaces at the expected level. It is evident that the selected chemical solutions and their percentages positively influenced the SBS values, although not to a statistically significant degree. Furthermore, it was observed that the SBS values increased with higher percentages. Additionally, clinically acceptable SBS values were noted in groups other than the Control and 4% HF acid groups, and these values exceeded the recommended 18 MPa (Zakavi et al., 2021).

The surface roughness values were statistically affected (p < 0.05) after the surface pretreatments were applied to the fiber posts. Surface roughness and SEM images indicate that chemical surface pretreatment applications at various concentrations can impact the fiber post surface in different ways.

This study tested only one resin cement and fiber post type under in vitro conditions, which limits the generalizability of the findings. Future studies should include multiple resin cements and post types to provide a broader understanding of the results. To enhance the investigation of bonding durability, thermocycling could be added as an aging parameter. Additionally, the long-term effects of the selected solutions on bond strength should be investigated. Changes in bond strength may occur due to aging, and the long-term effects of the solutions may vary. Furthermore, increasing the sample size in future studies could provide more robust statistical power and enhance the generalizability of the findings.

Conclusion

This preliminary study indicates that chemical surface pretreatments can be safely used to enhance the bond strength between fiber posts and resin cement, as they are not time-consuming, easy to use, and clinically applicable.

Supplemental Information

Supplemental Information 1 Raw data.

Additional Information and Declarations

Competing Interests

The authors declare that they have no competing interests.

Author Contributions

Elif Turkes Basaran conceived and designed the experiments, performed the experiments, analyzed the data, prepared figures and/or tables, authored or reviewed drafts of the article, and approved the final draft.

Magrur Kazak conceived and designed the experiments, analyzed the data, authored or reviewed drafts of the article, and approved the final draft.

Kagan Gokce performed the experiments, prepared figures and/or tables, and approved the final draft.

Yasemin Benderli Gokce conceived and designed the experiments, authored or reviewed drafts of the article, and approved the final draft.

Data Availability

The following information was supplied regarding data availability:

Raw data is available as a Supplemental File.

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
