# Peer review of "Effects of chemical surface pretreatments on the surface roughness and shear bond strength of fiber post-resin cement interfaces: a preliminary in vitro study"

_PeerJ, doi:10.7717/peerj.19853_

## Round 0.1 · original submission · Major Revisions

Please respond to the reviews.

**Language Note:** The review process has identified that the English language must be improved. PeerJ can provide language editing services - please contact us at [email protected] for pricing (be sure to provide your manuscript number and title). Alternatively, you should make your own arrangements to improve the language quality and provide details in your response letter. – PeerJ Staff

Reviewer 1 ·

Basic reporting

The manuscript is generally well-written, but minor grammatical revisions are suggested.

Figures 1–3 are illustrative but lack detailed captions. Include specific dimensions or experimental conditions in the captions.

Table 2 and Table 3 present statistical results clearly, but the p-values in Table 2 could be formatted consistently.

Experimental design

The power analysis is noted (n=10 per group), but the assumed effect size (1 unit) and standard deviation (0.5) should be justified with prior literature or pilot data.

The control group (no pretreatment) is appropriate, but consider clarifying why fibers were exposed during preparation (lines 269–270), as this may confound results.

The horizontal surface testing is innovative but warrants further discussion on how it mimics clinical scenarios compared to traditional top-surface testing.

Validity of the findings

The use of only one resin cement and post type limits generalizability. Acknowledge this and suggest future studies with multiple materials.


Aging was not included. Address how this affects the interpretation of bond durability.

Additional comments

The cohesive failure mode is noted, but include representative images or descriptive statistics to support this observation.

Reviewer 2 ·

Basic reporting

Thank you very much for providing the review article entitled “Effects of different surface pretreatments on fiber post-resin cement interface shear bond strength: a preliminary in vitro study”. The outline of the article is correctly structured and contains sufficient background information. Figures and tables are correctly positioned.

Experimental design

The experimental design is correctly outlined, but the methodology used is limited. The article would benefit from using more means of mechanical evaluation or imaging of the surface after treatment in terms of roughness or morphological changes - please consider expanding the article, as it only includes evaluation of SBS and optical microscopy.

Validity of the findings

Conclusions of the article: The findings of the article can be further evaluated for clinical applicability and modulation of SBS during clinical procedures.

Additional comments

1. Table 1 would benefit from a reference to where the material composition was found.
2. Table 2 should include the significance limit of the variable (p).
3. Please include the unit of SBS in Table 3.
4. Please label the items in Figure 1 and Figure 3. Figure 4 is missing the unit of SBS, and the legend is illegible due to the quality of the image - please revise.
5. In the conclusions, the authors state that "chemical surface pretreatments ... show no color change" - please comment on how this was assessed. There was no mention of any colorimetric method used.
6. Please consider including the name of the commercial material evaluated in the abstract.
7. Although light microscopy was mentioned in the methodology and discussion, no image is included in the article - please include relevant figures.

---

## Round 0.2 · Minor Revisions

Please address the remaining minor points.

Reviewer 1 ·

Basic reporting

No comment

Experimental design

No comment

Validity of the findings

No comment

Additional comments

The authors should also consider expanding the sample size or including aging protocols in future work.

Reviewer 2 ·

Basic reporting

Thank you very much for providing the revised version of the article now entitled “Effects of chemical surface pretreatments on the surface roughness and shear bond strength of fiber post-resin cement interfaces: A preliminary in vitro study”. The outline of the article is still correctly structured and contains sufficient background information. Figures and tables are correctly positioned.

Experimental design

The experimental design is correctly outlined. Thank you for reworking the methodology and including additional imaging and roughness assesment of the samples.

Validity of the findings

Thank you for reworking the discussion section in terms of clinical applicability and different scenarios.

Additional comments

I accept the responses and changes applied for comments 1-7 from previous round of review.

Additional comments:
1. Typo in Table 5 "Ssurface"
2. Figures 5-9 - the scale is not visible. Please rework legend for readability (colour and size).

---

## Round 0.3 · accepted · Accept

The manuscript can be accepted in its current form.

Reviewer 2 ·

Basic reporting

-

Experimental design

-

Validity of the findings

-

Additional comments

No additional comment in the light of previous rounds of review.